# Persistent Tracers of Historic Ice Flow in Glacial Stratigraphy near Kamb Ice Stream, West Antarctica

Nicholas Holschuh[1], Knut Christianson[1], Howard Conway[1], Robert W. Jacobel[2], Brian C. Welch[2]

[1]Department of Earth and Space Sciences, University of Washington, Johnson Hall Rm-070, Box 351310, 4000 15th Avenue NE, Seattle, Washington 98195-1310

[2]Department of Physics, St. Olaf College, 1520 St. Olaf Avenue, Northfield, MN 55057

*Correspondence to*: Nicholas Holschuh (holschuh@uw.edu)

**Abstract.** Variations in properties controlling ice flow (e.g., topography, accumulation rate, basal friction) are recorded by structures in glacial stratigraphy. When anomalies that disturb the stratigraphy are fixed in space, the structures they produce advect away from the source, and can be used to trace flow pathways and reconstruct ice-flow patterns of the past. Here we provide an example of one of these persistent tracers: a prominent unconformity in the glacial layering that originates at Mt. Resnik, part of a subglacial volcanic complex near Kamb Ice Stream in central West Antarctica. The unconformity records a change in the regional thinning behavior seemingly coincident (~3440 ± 117a) with stabilization of grounding-line retreat in the Ross Sea Embayment. We argue that this feature records both the flow and thinning history far upstream of the Ross Sea grounding line, indicating a limited influence of observed ice-stream stagnation cycles on large-scale ice-sheet routing over the last ~5700 years.

## 1 Introduction

New constraints on paleo-ice-dynamics are increasingly important in glaciology. They form the basis for validating model hindcasts, which act as a test of model performance, add to our understanding of past ice-sheet and climate interactions, and improve the reliability of future ice-sheet projections (Pollard et al., 2015). But current proxies for past ice-sheet behavior are not well distributed in time and space, limiting our ability to validate model behavior for critical regions of Antarctica and Greenland. Improving the temporal and spatial coverage for proxies in the ice-sheet interior is especially important in regions where significant flow reorganization is currently occurring, such as the Siple Coast, where centennial-scale internal variability (Catania et al., 2012) could easily be misinterpreted as externally forced, multi-millennial trends.

Past ice-sheet behavior is primarily inferred from three types of data: sea-level proxies (e.g. Galeotti et al., 2016; Raymo and Mitrovica, 2012), indicators of paleomorphology in currently deglaciated regions (e.g., The RAISED Consortium et al., 2014), and local or in-situ ice-sheet data. Each of these proxies has limitations. Sea-level data are powerful constraints on large-scale ice-mass changes but are limited in their ability to spatially resolve ice-dynamic changes. Deglaciated landscapes can spatially

and temporally resolve ice-sheet states (Anderson et al., 2017; Brook et al., 1995; Levy et al., 2017; Stone et al., 2003) but are limited to reconstructions of behavior outside the margins of the modern ice sheets. Studies using local or in-situ ice-sheet data are our only method for constraining changes in ice sheet morphology and behavior in the continental interior, but these rely on ice-core and borehole data that are logistically challenging to collect and spatially limited (Delmotte et al., 1999; Siddall et

al., 2012; Waddington et al., 2005). In this study, we advance the use of radar data as an ice-dynamic proxy, highlighting a class of englacial structures that record past ice-flow behavior in the continental interior.

Thermal, frictional, accumulation, and subglacial topographic anomalies all drive englacial structure formation (Holschuh et al., 2017). The resulting disturbances in the glacial stratigraphy have been used to infer ice-flow reorganization in the Ross Sea sector in the past (Conway et al., 2002; Jacobel et al., 1996; Siegert et al., 2004), but interpreting them can be challenging.

With uncertainty in both their formation mechanism and subsequent evolution, many of the englacial structures observed in Antarctica and Greenland do not provide sufficient information to infer paleo-velocities. However, temporally persistent thermal, frictional, and topographic anomalies (i.e. geologic controls) produce structures with unambiguous source locations, and like hotspot tracks on the ocean floor, the structures they produce can be used to infer the dynamics of the ice sheet through time. We call these structures "persistent tracers".

In this study, we focus on a persistent tracer in the catchment region of Kamb Ice Stream, West Antarctica (Fig. 1a) – a stratigraphic unconformity forming in response to Mt. Resnik, a high-relief, subglacial volcanic system upstream (Behrendt et al., 2006). This feature penetrates through over 1000m of ice, recording its formation and transport history for at least the last 6000 years. The flow history derived from this structure fills a gap in the proxy record for the Siple Coast, where flow stripes in the Ross Ice Shelf record the most recent history (~1000 a, Hulbe and Fahnestock (2007)) but records of the region's more

distant past are limited in space and time (Conway et al., 1999; Kingslake et al., 2018; Spector et al., 2017).

Outstanding questions exist regarding the role of individual ice streams in the overall mass budget of the Siple Coast from the Last Glacial Maximum (LGM) to today. The ice streams feeding the Ross Ice Shelf exhibit significant internal variability, with tidal forcing and frictional mechanics dictating velocity on hourly timescales (Anandakrishnan et al., 2003; Winberry et al., 2014) and thermodynamic changes resulting in significant local mass-balance variability on centennial timescales (Catania et

al., 2012). During the observational era, this variability has occurred primarily within the ice plains of Whillans and Kamb Ice Streams (Martín-Español et al., 2016), where low-relief subglacial topography and weak driving stresses allow subtle changes in the subglacial hydrologic system to significantly impact ice fluxes through individual ice streams (Siegfried et al., 2016). But the role of these ice-stream stagnation cycles in the region's overall ice-flow behavior is unknown. Previous studies have compared the Siple Coast to a braided stream, which can maintain a constant discharge despite rapidly changing flow pathways

(Fahnestock et al., 2000; Hulbe and Fahnestock, 2007; Parizek et al., 2002). In this study, we evaluate whether centennial-scale changes at the coast manifest in the ice reservoirs of the ice-sheet interior over the last several thousand years.

To do this, we use the 3D geometry of the Mt. Resnik unconformity to diagnose changes in ice flow and ice thickness. Mt. Resnik sits at the boundary between ice-stream catchments – structures forming here are well poised to capture substantial changes in relative flow between the ice streams to the south (Kamb, Whillans, and Mercer) and those discharging into the Ross Ice Shelf to the north (Bindschadler and MacAyeal). Like other persistent tracers seen elsewhere in Antarctica (Ross et al., 2011; Woodward and King, 2009), these data provide multi-millennial context for the ice-flow reorganization observed during the satellite era. We highlight this feature as one example of a larger class of structures that should be targeted in future radar studies of the Siple Coast Ice Streams.

## 2 Data

### 2.1 Radar Surveys

Ground-based radar campaigns conducted from the Byrd camp were performed in 2002 as part of the US-ITASE traverse using the St. Olaf College 3 MHz radar system (Welch and Jacobel, 2003) and in 2004 as part of the WAIS divide site surveys using the University of Washington deep-sounding 1 MHz radar system (Fig. 1B.i-ii). Radar data processing follows the workflow presented by Christianson et al. (2012). The ITASE data reveal a prominent unconformity within englacial layers in the top 1000m of the ice column (Fig. 2.D). The subsequent University of Washington survey shows the unconformity extends to Mt. Resnik, but is absent upstream of the topographic high (Fig. 2.B). System noise prevents interpretation of layers shallower than ~250m depth.

### 2.2 Byrd Core and Reflector Chronology

Radar data from the ITASE traverse connect to the deep ice core retrieved at Byrd station in 1968. Damaged and missing core precluded the counting of annual layers, but a shallow depth-age scale was established using the electrical conductivity method (Hammer et al., 1994). This overlaps with a chronology starting at 870m depth, which maps ages from the well dated Greenland ice cores to Byrd via methane correlation (Blunier and Brook, 2001). Following the methods of Cavitte et al. (2016), we date radar reflectors where they intersect the Byrd ice core, taking into account both published uncertainty values and the magnitude of disagreement between conductivity and methane inferred ages where they overlap. The age and uncertainties of radar layers (presented as years before 2000 A.D. [a]) are given in Figure 2.C. Note the presence of an unconformity, a gently dipping feature with two breaks in slope that crosscuts otherwise continuous internal layers. We refer to its geometry using the dated layers that truncate nearest to the slope breaks, one near the $1648 \pm 92$a reflector, and one between the $3011 \pm 115$a and $3440 \pm 117$a reflectors.

Dating reflectors across the unconformity is challenging, as no radar lines can tie reflectors on the far side of the unconformity through the undisturbed layering upstream of Mt. Resnik. Common reflections were correlated based on their absolute

amplitude and their amplitude and waveform characteristics relative to the adjacent reflectors in the stratigraphy. While we have high confidence in the layer correlation, we mark the reflectors not directly linked to the ice core with dotted lines (Fig. 2.D).

## 2.3 Satellite Imagery

Englacial unconformities are typically associated with wind-scour and sublimation (Das et al., 2013), often the result of steep surface gradients. Two different methods have been used to identify surface scour in satellite data over Mt. Resnik. Bright reflectivity in MODIS imagery (like that seen in Fig. 3.A) indicates local grain-size reduction (Scambos et al., 2007), consistent with unconformity generation elsewhere (Welch and Jacobel, 2005). Additionally, calibrated studies of surface spectral properties show that blue-ice with bubbles and snow have comparable reflectance in the visible spectrum, but blue ice is a

substantially weaker reflector in the near-infrared (Boresjö Bronge and Bronge, 1999). Using bands 2 (452-512 nm) and 5 (851-579 nm) of Landsat 8 data collected over Mt. Resnik (Path 224, Row 119, acquired Jan 21, 2018), we examine the relative surface reflectivities at the location of high MODIS reflectivity (Fig. 3.B-C). With a threshold reflectivity ratio of 2, we identify likely blue ice areas, plotted in Figure 3.C. These correspond with peak elevations in the subglacial topography, and fall along a roughly linear ridge orthogonal to flow. Based on the modern flow field, the downstream unconformity position is roughly

consistent with formation at the edge of the blue ice patch observed over Mt. Resnik (Fig. 3.A).

## 3 Results and Discussion

Unconformities manifest at the boundary between depositional regimes at the ice-sheet surface. They can be defined by substantial missing time, due to erosion/ablation or non-deposition in blue ice areas, or can represent the transition from pristine to mechanically reworked snow surfaces, as in the case of the megadunes of East Antarctica (Frezzotti et al., 2002). For an

unconformity to appear in radar imagery, surface processes must modify the depth-conductivity profile – snow from one regime must sit on top of snow from another regime.

Downstream of a blue ice area, radar data collected along flow should capture an unconformity that dips away from the zone of surface ablation. This would mark the transition between snow that predates and postdates the missing time in the column. But there may be no indication of an unconformity in radar data collected orthogonal to flow, as the missing time would appear

like any other isochrone in the layering. This highlights an important feature of the unconformity seen in radar data downstream from Mt. Resnik: its geometry is quite complex in the cross-flow direction. If it were forming in a simple flow field, from a single blue ice area, we would not expect a sloping feature like the one observed. But under certain temporally-stable ice-flow conditions, or in the presence of certain dynamic flow changes, the observed geometry can be explained. Here we analyze the

structure in detail, focused on two primary components: (1) its trace, as it advects away from Mt. Resnik, and (2) its cross-sectional geometry, which has the potential to constrain more subtle changes in ice dynamics around Mt. Resnik through time.

## 3.1 Propagation pathway from Mt. Resnik

The formation of the unconformity is unambiguously connected to Mt. Resnik. This is evident from the radar data; the unconformity is absent in data collected upstream of Mt. Resnik, and is visible in all downstream lines (see Fig 2.B, and Supplementary Fig. 1). Enhanced driving stress is required to drive ice flow around and over the mountain (in the < 400m of ice that flows over its summit (Morse et al., 2002)), resulting in steep surface gradients observed in the altimetry data (discussed in Section 3.3), and driving surface scour observed in the imagery.

Despite significant variability in the configuration of the modern Siple Coast ice streams, there is very little evidence in the trace of the Mt Resnik unconformity that ice-flow direction here has changed significantly in the last ~5700 years. Its current propagation direction is roughly coincident with the flow paths predicted from the modern velocity field, shown in Figure 1.C. This is surprising given the recent shutdown of Kamb Ice Stream (Anandakrishnan and Alley, 1997), whose tributary dominates ice flow immediately south of Mt. Resnik. The trace of the unconformity indicates that transitions between stagnant and active flow of the Kamb Ice Stream must not have significantly modified the direction of driving stresses in the ice-sheet interior, as ice flowing off Mt. Resnik has not been diverted in-to or out-of the neighboring tributaries.

## 3.2 Formation Mechanisms and Geometry Interpretation

This unconformity is a time-transgressive structure; it is present across a range of depths (and therefore, a range of ages) within a single cross section. There are a limited number of scenarios that are consistent with this structure – special configurations of the erosive anomaly and the velocity field can form a time-transgressive unconformity in steady state, or the erosion area and/or flow field can change with time. We describe the steady-state and transient mechanisms in Figure 4, and discuss our favored mechanism and how it can further inform our understanding of ice dynamics near Mt. Resnik.

The first steady-state mechanism that could generate the observed unconformity (Fig. 4a) relies on an erosive anomaly at the surface that mirrors the kinked shape of the unconformity in the subsurface. Assuming that ice flows away from the source region and is buried at a constant rate, the deeper components of the unconformity must have formed further away than the shallowest limb. Based on the data presented in Figure 2, we know that the formation period of the unconformity spans at least 3000 years. Given surface velocities of ~5 m/a (consistent with InSAR-derived velocities for the region), this mechanism would require a formation area that extends 15km in the along flow direction to span 3000 years of stratigraphy. Because the blue ice fields seen in Landsat imagery (Fig. 3c) appear along a roughly linear trace, orthogonal to flow, this formation mechanism is unlikely.

The alternative steady-state mechanism produces the observed unconformity through lateral velocity gradients (Fig. 4b), as variations in the velocity field are the only remaining characteristic of the system capable of producing the range of transit times (and associated burial depths) required for a sloping structure. The unconformity is shallower to the true south (grid north), indicative of a shorter transit time since formation. This would be consistent with the faster flow observed in the active Kamb tributary that bounds Mt. Resnik to the south. The variation in unconformity depth within the profile is also reasonable given the range of observed surface velocities, with ice flowing a factor of ~3 faster within the Kamb tributary. However, to reproduce the exact geometry of the unconformity, the lateral acceleration in flow speed must be localized in a narrow (~5km) shear margin, with two breaks in slope in the velocity gradients corresponding with kinks in the unconformity. These features are not present in the modern velocity field.

Transient formation mechanisms could take several forms, but the simplest mechanism for the observed geometry (Fig. 4c) relies on an erosive area that changes shape through time. The sloping, kinked unconformity would form if the boundary defining the southern edge of the Mt. Resnik blue ice area (and as a result, the area downstream of the blue ice, where deposition is perturbed) migrated southward in time, with punctuated changes in the migration rate corresponding to the breaks in slope the unconformity. This way, as the erosive area expands, the boundary between modified deposits downstream of the blue ice field and snow deposited under more quiescent conditions would slope up toward the surface in the direction of blue ice expansion. The northern boundary, however, must stay fixed, leading to no unconformity visible in the flow-orthogonal radar data on that side.

In the case of Mt. Resnik, if the ice sheet were to progressively thin, the subglacial topography would exert more local control on the surface gradients, and the blue ice area would expand. Steep slopes and smooth surfaces over blue ice enhance katabatic winds, and can ultimately drive turbulence that reworks snow in the depositional areas downstream. This process has been seen elsewhere in Antarctica, and was highlighted in Figure 4 of Bintanja (1999), who show that changing ice thickness in the vicinity of rugged subglacial topography will induce changes in the local depositional regime.

The kink locations within the unconformity provide additional information about ice dynamics in the region. The kink can either be explained by the spatial characteristics of the formation mechanism (as in mechanisms 1 and 2), or in the temporal characteristics of formation (as in mechanism 3). Without any evidence for a kinked structure in either the velocity field or the blue ice patches observed, we use the dated reflectors to estimate what ages would be associated with the kinks, should they be the result of temporal variability in formation. The deeper of two kinks in the unconformity falls between reflectors dated at ~3.4ka and 3.0ka. This is contemporaneous with the activation (Conway et al., 1999) and migration of divide flow in the region (Nereson and Raymond, 2001), rapid grounding-line retreat to its present position in the south western Ross Embayment (Spector et al., 2017), and a change in the thinning behavior in western Marie Byrd Land (Stone et al., 2003). Interior thinning at Mt. Resnik is likely to occur with grounding-line retreat, consistent with a southward migration of the edge of the erosional zone driven by changes in local ice thickness over Mt. Resnik.

Under our preferred interpretation, that the kinks represent temporal changes in ice-flow behavior, it is possible to interpret the other geometric features of the unconformity. Between 3.2ka and 1.1ka, there is a limited record of dynamic change in the region from other paleo-flow proxies. This means that the second transition, corresponding to a slope break at ~1.65ka, would represent a previously undocumented transition in thinning rates upstream of the Siple Coast. This may simply be the temporal

lag between grounding-line stabilization around 3ka and the resulting equilibration in the ice-sheet interior, but it highlights the possibility of ice-dynamics changes that have not yet been described. Other records of ice-sheet behavior post-dating 1.65ka are consistent with continued southward migration of the scour surface at Mt. Resnik, as thinning of the southern Siple Coast ice streams continued through the past ~1000 years (Nereson and Raymond, 2001).

## 3.4 Implications for unconformities elsewhere in Antarctica and Greenland

Unconformities are rare in West Antarctica, facilitated here by the extreme subglacial relief of Mt. Resnik and its influence on local surface gradients. Previous studies have related observed wind scour to two basic surface parameters: the ratio of the surface mass balance to the average wind speed (A/W: $[kgm^{-2}a^{-1}]/[ms^{-1}]$), and the mean surface slope in the wind direction (MSWD) (Das et al., 2015). This empirical framework predicts wind scour in regions where A/W < 9.12 and MSWD > 0.002, with thresholds established using observations from Dome A in East Antarctica. However, the Dome A training data spans a

relatively narrow range of surface slopes and accumulation rates, not sampling the values expected over Mt. Resnik. The presence of blue ice at Mt. Resnik indicates an expanded range of conditions that allow scour, which we can identify using modeled accumulation rates, wind speed, and observed surface elevation co-located with our radar data.

We compute the A/W ratio and the MSWD over Mt. Resnik using output from a regional climate model at ~ 30km resolution (RACMO (Noël et al., 2015)) and 8m resolution digital elevation models (DEMs - produced from orthoimagery collected by

the DigitalGlobe constellation of satellites, using the SETSM algorithm (Noh and Howat, 2015)). ICESat data (version 34, GLAH12) were used to remove errors in regional slope in the DEMs. While there is a temporal gap between the DigitalGlobe (01/2015-12/2016) and ICESat data acquisitions (02/2003-10/2009), we do not expect significant changes in regional slope over this time period, given small observed dh/dt signals here (Helm et al., 2014). The coverage of these regional DEMs is provided in Figure 3.A.

The A/W ratio for this region is significantly higher than the values reported for scour regions around Dome A. 365-day averages from 2000 to 2009 show A/W ratios oscillating around 20.2, more than a factor of 2 higher than the threshold. However, high surface slopes are also found over Mt. Resnik, with MSWD values in both published DEMs from Cryosat [MSWD = 0.018] (Helm et al., 2014) and the regional DEMs produced for this study [MSWD = 0.0192 ± 0.002] exceeding four times those at Dome A. Because the resolution mismatch between the atmospheric forcing and the DEM likely results in

underestimates of the MSWD (as topographic focusing strengthens surface winds locally), these values provide a lower bound on wind speeds capable of producing unconformities in high accumulation regions of Antarctica and Greenland.

## 4 Conclusions

Mt. Resnik produces an unconformity in the glacial stratigraphy of central West Antarctica that acts as a persistent tracer for ice flow through the Kamb / Bindschadler Ice Stream systems. The trace of the unconformity indicates no gross changes in ice-flow direction in the Kamb Ice Stream catchment over the last ~5700 years, despite the dramatic changes in coastal flow regime observed over the satellite era and inferred from flow-striping on the Ross Ice Shelf. Thus, we believe that the response time for the ice-sheet interior exceeds the stagnation-activation time scales for the Kamb Ice Stream system, resulting in damped coastal forcing at Mt. Resnik and a record of only long-term average behavior. This could imply a limited effect of ice-stream stagnation-reactivation cycles on the total ice flux to the ocean.

If our preferred mechanism for unconformity formation is correct, the structure records a thinning trend in the ice sheet interior, with punctuated accelerations in thinning at two distinct events. The first (between ~3.4ka and 3.0ka) corresponds with a known change in ground-line behavior, but the second (1648 ± 92a) is a previously undocumented event in the ice-dynamic history of the Siple Coast. Further data collection is required to identify the source of the more recent change in behavior.

This study provides historical context for modern changes observed at Kamb Ice Stream, identifies a new range of environmental conditions that permit surface erosion in Antarctica and Greenland, and highlights the utility of persistent tracers in paleo-ice-flow reconstruction. Persistent tracers like the one described at Mt. Resnik are actively forming where there are spatially locked boundary forcing anomalies, and their locations can be predicted from surface observations, bed topographies, and model-parameter inversions. We provide a single example, but encourage future studies to seek these structures out explicitly. For regions where other paleo-ice-dynamic proxies are difficult to collect, persistent tracers might fill both the spatial and temporal gaps that limit our ability to constrain hindcasts of ice-sheet behavior.

## Data Availability

Radar data are accessible through the University of Washington's ResearchWorks Archive (http://hdl.handle.net/1773/42567).

## Acknowledgments

This work has been substantially improved through comments from our editor and reviewers, including several anonymous reviewers and Dr. Neil Ross. The Polar Geospatial Center at the University of MN provided support through their DigitalGlobe derived Antarctic DEMs. We thank the US Antarctic Program, Raytheon Polar Services, and 109th Airlift Wing of the NY Air National Guard for logistic support in collecting the radar data. This work was funded as a part of NASA grant NNX16AM01G, NSF grants OPP-9814574, and OPP-0087345.

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

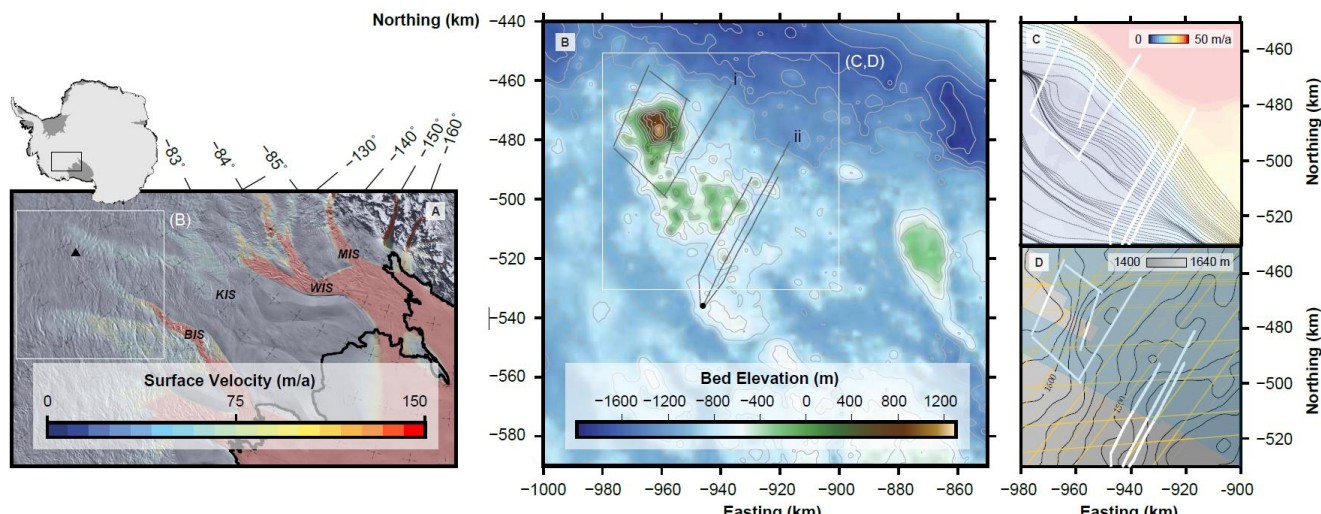

**Figure 1:** (A) Region map showing the modern flow field (Joughin and Tulaczyk, 2002) of Bindschadler (BIS), Kamb (KIS), Whillans (WIS), and Mercer (MIS) ice streams. Mt. Resnik, a subglacial volcanic complex (plotted as a black triangle), sits adjacent to a tributary of the stagnating Kamb Ice Stream, near the catchment divide between Kamb and Bindschadler Ice Streams. (B) Map of the subglacial topography at Mt. Resnik (Morse et al., 2002). Two ground-based radar surveys are plotted in black: (i) 1 MHz data collected by the University of Washington in 2004, and (ii) 3 MHz data collected by St. Olaf College in 2002. (C) Map of modern flow speeds and flow pathways over Mt. Resnik (dashed lines) (Joughin and Tulaczyk, 2002), indicating a dominant flow direction orthogonal to the primary radar survey orientation. (D) Contoured [20m] surface elevation (Helm et al., 2014), plotted with the coverage region for high resolution DEMs produced as part of this study (blue) calibrated using ICESat altimetry (ground-tracks in orange, data version 34, GLAH12).

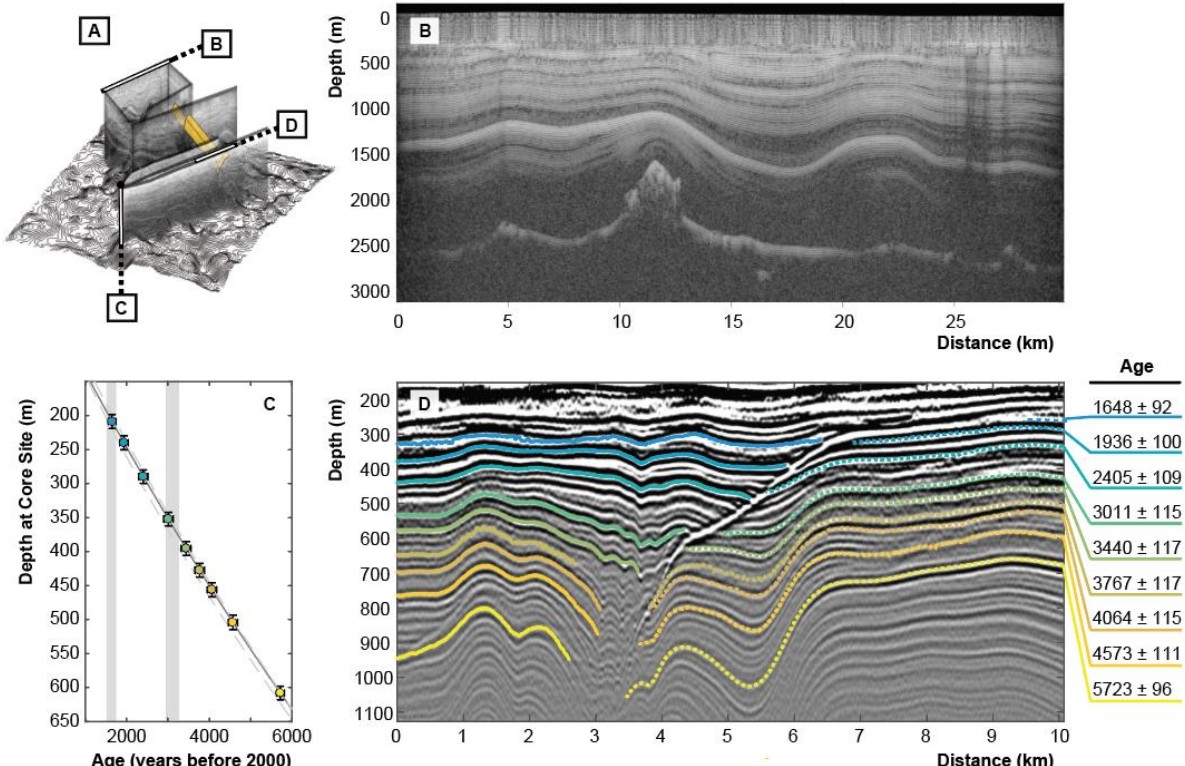

**Figure 2**: (A) Fence diagram, indicating the positions of the upstream radar survey (B), the Byrd Ice Core (C), and an example downstream line containing the unconformity of interest (D). (B) The radar profile immediately upstream of Mt. Resnik, highlighting conformable layering. (C) Dated reflectors and their associated uncertainties, plotted as a function of depth at the Byrd ice-core site. Grey bars indicate dated slope breaks in the unconformity, potential indications of historic ice-dynamic changes (discussed in section 3.2). (D) Dated reflectors traced on the 3 MHz ITASE radargram, with ages (in years before 2000 A.D.) labelled. Dotted lines indicate reflectors dated by amplitude and waveform correlation across the unconformity, solid lines are traced continuously from the ice-core site.

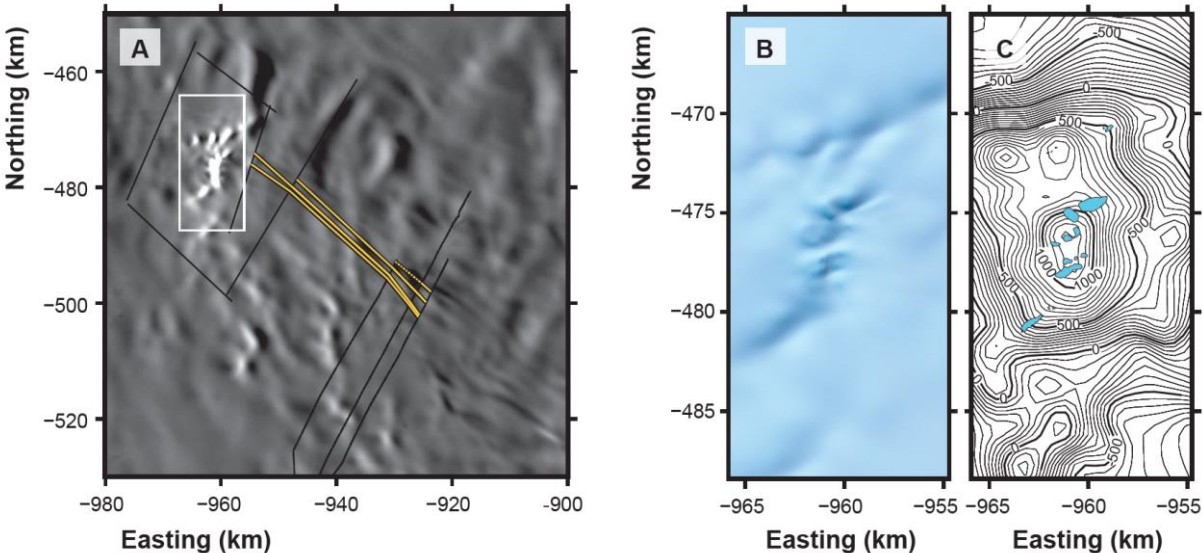

**Figure 3**: (A) MODIS Imagery for the region (Haran et al., 2014), showing high reflectivity at the source of the unconformity over Mt. Resnik. Traces of slope-breaks in the unconformity are plotted in yellow. (B) False-color Landsat 8 imagery collected over Mt. Resnik, using the near-infrared (band 5), green (band 3), and blue (band 2). (C) Contoured basal topography, with blue-ice areas (inferred from blue to near-infrared reflection intensity ratios > 2) highlighted.

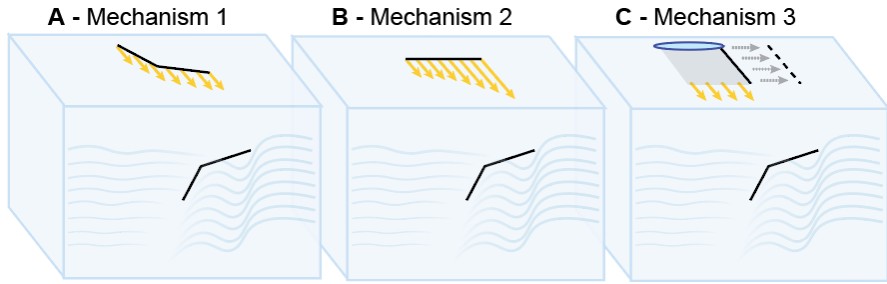

**Figure 4:** Schematic detailing end-member mechanisms for unconformity formation with the distinctive, kinked geometry observed in the radar data: (1) a stationary surface feature orthogonal to flow that mirrors the shape of the unconformity in the subsurface, advecting away and buried with constant velocities, (2) a stationary surface feature orthogonal to flow that is advected and buried in a spatially variable velocity field, with velocity gradients that mirror depth gradients of the structure, or (3), a feature whose boundaries drift with time, with the slope of the unconformity varying as a function of the rate of drift. We believe that mechanism 3 is most consistent with the unconformity geometry, with kink positions corresponding to the dates of ice-dynamic changes in the region.

