# Peer review of "Persistent Tracers of Historic Ice Flow in Glacial Stratigraphy near Kamb Ice Stream, West Antarctica"

_The Cryosphere, 2018_

## Referee Comment (RC1) · Anonymous Referee #1 · 11 Jun 2018

General comments

Here I review "Persistent Tracers of Historic Ice Flow in Glacial Stratigraphy near Kamb Ice Stream, West Antarctica" by Holschuh et al. submitted to The Cryosphere Discussions. In this paper the authors use early 2000s ground-based radar profiles from the Siple Coast to image the radio-stratigraphy of an area near Kamb and Bindschadler Ice Stream onset areas.

The authors find a discontinuity in the radio-stratigraphy on the lee side of Mt. Resnik, which the paper discusses possible causes of. As I understand the paper, the authors favor a mechanism in which the thinning of ice over Mt Resnik after 3.4 ka leads to

changes in surface slope. In turn, this drives a southern expansion of the blue ice area in the lee of the subglacial obstacle. This erosive region then explains both the presence of an unconformity and its kinked morphology.

I enjoyed the paper and think it makes a worthwhile contribution to the glaciological understanding of the locality and the wider region. Broadly the scientific content of this paper is good, and the conclusions are valid given the presented data. However, the progression of the paper is often difficult to follow and requires major reorganisation before I can recommend it for publication. I encourage a resubmission and I would be happy to review a revised manuscript.

Specific Comments

I found the introduction section confusing, with an "Introduction (1.1)" section and then two subsidiary sections (1.2 and 1.3). Section 1.1 states what a persistent tracer is and the direction of the paper, before repeating and expanding this in the subsequent Introduction sections. The content is there but needs reorganising to a clearer progression from: Scientific background, what englacial tracers can contribute, and how these will be applied to the study area to work towards the paper's scientific and methodological conclusions.

The need for Section 2.3 is not immediately clear to me and appears somewhat surprisingly after no mention of this line of evidence in the Introduction. I would argue that much of this section is not "Data" at all but analysis performed to explain the observations in the data presented in Sections 2.1 and 2.2, and it must therefore be integrated into Section 3 or a Methods section.

In 2.3 the authors are seeking to explore reasons why the radar unconformity is present using RACMO model output at much lower resolution than the target features. This line of evidence and enquiry is then abandoned, and used to motivate Section 2.4 which looks for the occurrence of surface scour in satellite imagery. It would be more logical to first see if a blue ice area exists, and then try to explain its occurence using climate

model output.

The beginning of Section "3 Results" is not Results. P5 L13-25 are Introduction (that Mt Resnik is a good tracer, what layers record in general, current ideas about what unconformities represent), and most of P5 L24-P6 L2 should be integrated into Section 3.1. In Section 3.1 the idea that Mt Resnik is a persistent tracer is unnecessarily repeated again (P6 L8-9).

Another structural point arises in Section 3.2 which I found a little confusing. At P6 L27-30 the authors reject the first formation mechanism based on some travel time calculations. The paper then goes onto "select a favored formation mechanism" in Section 3.3. having already rejected one of the mechanisms. This dis-order also happens for the second mechanism (P7 L5-7).

Technical comments

P1 L27. Ross et al., 2011 not in reference list?

P2 L9. Suggest a rewrite of "limited to reconstructions of behaviour *outboard* of current margins" as meaning not immediately obvious.

P2 L24. Stylistic point - I think "centennial timescales" is more standard than "century timescales"

P3 L15-21. Check whether the in text referencing to Figures is correct. Text states Fig 2.B shows unconformity but caption and image itself show conformable layers. 2.A is similarly contradictory.

P6 P24. It would be clearer is the specific figure panel was referred to in text – e.g. Fig 4a, etc.

P7 L1-7. Is Mechanism 2 mutually exclusive from the other mechanisms?

P7 L12-14. The meaning of the sentence "This way, snow....in the radar data" is unclear. Does the blue ice area explain the unconformity? Or does the change from

"quiescent" or "turbulent" snow deposition?

P8 L12-13. I wouldn't say basal friction, accumulation or melt are "spatially locked" as all three are emergent and dynamic boundary conditions in ice sheet flow and, therefore, stratigraphy.

Fig 3. It would be informative to see an indication of the radar unconformity zone and modern ice flow mapped onto the current Landsat-8 identified modern day blue ice areas.

Fig S1. Caption L4. "un" typo.

[Figure]

---

## Referee Comment (RC2) · Anonymous Referee #2 · 6 Jul 2018

This paper is concerned with observations of an unconformity in englacial layers in the onset region of Siple Coast Ice Streams and their interpretation in the context of the ice flow history of that region. To my understanding the analysis is sound, and the interpretation provided by the authors is supported by the observations presented.

My two main points (detailed below) are relatively minor, and concern primarily the organization of the paper and the writing style. Overall, I find that the manuscript is solid, and I support publication provided my comments are addressed.

Main points:

1) It seems to me that the conclusion that the tributaries of the Siple Coast Ice Streams

have remained stable in direction during periods of sustained grounding line migration is an important result, which is overlooked throughout the paper and very briefly discussed at the end of the Conclusions sections. Given that very little is known about the dynamics of ice streams and their tributaries, I would encourage the authors to stress this conclusion and to make it front and centre of the Results and Conclusions sections.

2) The text in very succinct, at times to the point that it is hard to follow (a few instances are indicated in the minor points). I would recommend that the authors revise the text in this light. Further, I find that the figures are somewhat disconnected from the text, while they could be used to support it and clarify the writing in a much more effective way.

Minor points:

- page 5, line 29: what do the authors mean with "static flow fields"? Steady (no change in time), perhaps?

- page 6, line 12: Figure 3C is not the right figure

- page 6, line 27-30: here you use the present-day configuration of the blue ice region to reject one formation mechanism, but it's unclear to me how/ under what assumptions this applies to the past. Can you expand on this?

- page 7, lines 1-7: in my opinion, this paragraph is barely understandable. I recommend that the same description is rewritten with closer reference to the supporting figure, and disentangling interpretation from observations. Also, the notation " 3x, 5x, .. " is highly confusing

- page 7, line 17: " Steep slopes .. " it might be obvious, but I would briefly explain why steep slopes over blue ice enhance the winds

- figure 1d: what is the colour scale?

[Figure]

[Figure]

---

## Author Response (AR1)

Dear Dr. Karlsson:

We would like to thank our two anonymous reviewers for their comments on our manuscript, "Persistent Tracers of Historic Ice Flow in Glacial Stratigraphy near Kamb Ice Stream, West Antarctica." Both reviewers found the work interesting, and the science compelling. Thus, their comments focused exclusively on manuscript clarity, indicating that slight changes to structure and style might make the work more accessible to future readers. By eliminating several section headings, modifying our description of the erosional mechanisms, and reorganizing the introduction, we believe we have improved the manuscript and addressed all of the reviewer concerns.

Below, we provide our responses to each reviewer, which detail the changes made in response to the individual comment made. If you would like more information about any changes, please do not hesitate to ask.

Thank you very much for your consideration,

Nick Holschuh, Knut Christianson, Howard Conway, Robert Jacobel, and Brian Welch

Dear Reviewer 1:

Thank you for your thorough evaluation of our manuscript. Your comments on the paper's structure were appreciated and have ultimately led to a clearer narrative. Below we outline the organizational changes made in response to your review, and provide point-by-point response to your technical suggestions.

**1) Changes to the Introduction**

The original introduction was designed to separately emphasize the work's contributions to ice-sheet reconstruction methods as well as its contribution to the specific paleo-proxy record for the Siple Coast. We understand how that structure could seem fractured, and include material that was redundant, so we have eliminated the section headings and reorganized the introduction to provide the clearer progression you requested. Now, we introduce current paleo proxies (P1 L18 – P2 L6), indicate how englacial structures can act as paleo-flow proxies (P2 L7 – P2 L13), and finally discuss how persistent tracers could help us understand the complex Siple Coast Ice Streams (P2 L14 – P3 L8). We believe this fits your recommended structure of (1) scientific background, (2) what englacial tracers can contribute, and (3) how they will be applied to our study area.

**2) Section 2.3 – Environmental conditions required for erosion**

We never viewed this section as a line of evidence, per se, for the formation mechanism of the unconformity. We simply thought it was necessary to address the existing literature on unconformity formation, and provide context for how surface scour may be possible over Mt. Resnik despite the high accumulation rates in West Antarctica. For a more logical structure, we have moved the original section 2.3 into the discussion section of the paper (P7 L9 – P7 L31).

**3) Section 3 – Results**

We agree that the first paragraph of section 3 feels out of place – much of the content has been cut, and what is remains has been integrated into the introduction. However, much of the discussion of unconformity geometry would lack context if it were introduced before the data. As a result, we chose to keep what could be considered an "introduction to unconformities" here, but we have renamed section 3

to "Results and Discussion". We have also eliminated the section division between "possible mechanisms" and "favored mechanism", and adjusted phrasing in text to eliminate any repeated rejection of the disfavored mechanisms.

**Technical comments:**

*P1 L27. Ross et al., 2011 not in reference list?*
On our version, Ross et al. 2011 was included in the original reference list, P12 L17.

*P2 L9. Suggest a rewrite of "limited to reconstructions of behaviour \*outboard\* of current margins" as meaning not immediately obvious.*
Rewritten to clarify that we mean outside the ice-sheet margins (P2 L2).

*P2 L24. Stylistic point - I think "centennial timescales" is more standard than "century timescales"*
We have updated this to centennial.

*P3 L15-21. Check whether the in text referencing to Figures is correct. Text states Fig 2.B shows unconformity but caption and image itself show conformable layers. 2.A is similarly contradictory.*
This error reflected a previous version of the figure, which had since been updated to include additional panels. We have fixed these references in the current version of the manuscript.

*P6 P24. It would be clearer is the specific figure panel was referred to in text – e.g. Fig 4a, etc.*
Fixed.

*P7 L1-7. Is Mechanism 2 mutually exclusive from the other mechanisms?*
We do not view any of the mechanisms as mutually exclusive, but rather end-member scenarios that could result in the unconformity geometry. We have changed the text to better reflect this. Mechanisms 1 and 2 allow the formation of a time-transgressive, kinked unconformity under steady-state conditions, unlike mechanism 3 which is formation by non-steady forcing. Ultimately, we find the steady-state mechanisms unsatisfactory, thus, it must form by some evolution of surface processes.

*P7 L12-14. The meaning of the sentence "This way, snow....in the radar data" is unclear. Does the blue ice area explain the unconformity? Or does the change from "quiescent" or "turbulent" snow deposition?*
In order for this time-transgressive feature to exist, the formation zone must extend from near Mt. Resnik (where the ~6000 year old portion of it formed) to an area nearer to where the unconformity is observed in our data (where the ~1000 year old portion formed). Thus, the observed blue ice cannot be the entire area modifying the stratigraphy. We hypothesize that surface reworking downstream of the blue ice area also disturbs the depth-conductivity profile. As the blue ice area (and its downstream expression) expand, areas formerly unaffected by the blue ice zone are now being disturbed, leading to a conductivity contrast associated with that time of transition, and an imageable unconformity. We have made subtle changes to the text on P6 L12-15 to hopefully make this clearer.

*P8 L12-13. I wouldn't say basal friction, accumulation or melt are "spatially locked" as all three are emergent and dynamic boundary conditions in ice sheet flow and, therefore, stratigraphy.*
We did not intend to imply that any of these properties are inherently spatially locked, but that, in the event they are spatially locked, they result in a persistent tracer. We simply mean that long lived features

exist that affect the ice-sheet system through each of these pathways (subglacial topography, subglacial geologic features like sedimentary basins, or even "dynamic" features like lake Vostok which perturb the surface elevation and resulting accumulation field).

*Fig 3. It would be informative to see an indication of the radar unconformity zone and modern ice flow mapped onto the current Landsat-8 identified modern day blue ice areas.*
One thing we struggled with in making the figures for this paper is providing all the relevant coincident data sets. Currently, Figure 3 does provide the unconformity zone relative to the blue ice area – the only thing missing from your request is the modern flow field. We did reconstruct this figure including the flow paths plotted in Figure 1c, but felt it was cluttered and harder to interpret. As a result, we decided that it is best to leave this figure as is, and rely on Fig 3a and Fig1c together to provide the reader with enough context to understand the system.

*Fig S1. Caption L4. "un" typo.*
Fixed

Dear Reviewer 2,

Thank you for your comments on the text of our manuscript. The changes we made in response have led to a substantially improved work, most notably a clearer set of descriptions for the unconformity formation mechanisms. We outline those changes below, and provide point-by-point responses to your technical comments.

**1) Appropriately emphasize conclusions regarding the Siple Coast Ice Streams**

We are glad you find our results compelling, and we have restructured our conclusions to immediately discuss the long-term behavior of the Siple Coast Ice Streams. We drafted alternative structures of the results section, but found that the clearest structure starts with the mechanistic break down of unconformity formation, and follows with discussion of paleo-flow behavior. Hopefully the new conclusion provides the emphasis you were seeking.

**2) The text (most notably, the discussion of erosion formation mechanisms) is at times hard to follow.**

We have overhauled the text related to unconformity formation, in an effort to respond to your technical comments and better leverage the figure. We took special care to better explain mechanism 2 (now on P6, L1-8). Additional structural changes were made to accommodate your review and the other anonymous reviewer (see our response to review 1 for more details).

**Technical Comments:**

*page 5, line 29: what do the authors mean with "static flow fields"? Steady (no change in time), perhaps?*
[Now Page 4, Line 27] We have rephrased for clarity. "… temporally-stable ice-flow conditions …"

*- page 6, line 12: Figure 3C is not the right figure*
Fixed

*- page 6, line 27-30: here you use the present-day configuration of the blue ice region to reject one formation mechanism, but it's unclear to me how/ under what assumptions this applies to the past. Can you expand on this?*
We have made substantial changes to the text to make the distinction between mechanisms clear. For mechanisms 1 and 2, we are focused on formation processes that can explain the unconformity as a steady-state structure. In that sense, we are assuming that the configuration of the blue ice area is constant through time. Mechanism 3, our preferred mechanism, relies on changes to the configuration of the blue ice area through time to explain the unconformity. It is this distinction, steady-state versus transient, that makes mechanism 1 different from mechanism 3.

*- page 7, lines 1-7: in my opinion, this paragraph is barely understandable. I recommend that the same description is rewritten with closer reference to the supporting figure, and disentangling interpretation from observations. Also, the notation " 3x, 5x, .. " is highly confusing.*
This text has been rewritten to make the paragraph clearer, as well as eliminate the confusing nomenclature.

*- page 7, line 17: " Steep slopes .. " it might be obvious, but I would briefly explain why steep slopes over blue ice enhance the winds.*
We clarified this to read katabatic winds, as the surface slope primarily affects gravity driven flows.

*- figure 1d: what is the colour scale?*
Color scale added.

[revised manuscript text omitted]

15    We compute the A/W ratio and the MSWD over Mt. Resnik using output from a regional climate model at ~ 30km resolution (RACMO (Noël et al., 2015)) and 8m resolution digital elevation models (DEMs - produced from orthoimagery collected by the DigitalGlobe constellation of satellites, using the SETSM algorithm (Noh and Howat, 2015)). ICESat data were used to remove errors in regional slope in the DEMs. While there is a temporal gap between the DigitalGlobe (01/2015-12/2016) and ICESat data acquisitions (02/2003-10/2009), we do not expect significant changes in regional slope over this time period,
20    given small observed dh/dt signals here (Helm et al., 2014). The coverage of these regional DEMs is provided in Figure 3.A. We also note here the resolution mismatch between the atmospheric forcing and the DEM - we capture local slopes, but only regional surface mass balance and wind speed.

[revised manuscript text omitted]

**4 Conclusions**

5    Here we provide evidence for the existence of persistent tracers of historic ice flow within the Antarctic ice sheet. Like hot spot tracks on the ocean floor, these form in response to a spatially locked forcing (either by topography, basal friction, accumulation, melt, or some other boundary condition to the ice-flow equations) and propagate away from their source, recording the flow vector for the ice sheet in that process. The location of persistent tracers within the ice sheet can be predicted from both model inversions and from preliminary sparse data, so future field expeditions should seek them out as proxies for changes in flow behavior in our most dynamic or poorly understood regions of Antarctica and Greenland.

Mt. Resnik produces an unconformity in the glacial stratigraphy in central West Antarctica that acts as a persistent tracer for ice flow through the Kamb / Bindschadler Ice Stream systems. The trace of the unconformity indicates no gross changes in ice-flow direction in the Siple Coast catchment over the last ~5700 years recorded in the stratigraphy, despite dramatic changes in flow regime for more coastal regions both observed during the satellite era and inferred from flow-striping on the Ross Ice

15   Shelf. Thus, we believe these data imply that the response time for the ice-sheet interior exceeds the stagnation-activation time scales for the Kamb Ice Stream system, damping the signal and recording only long-term average behavior.

Detailed interpretation of the slope breaks in the Resnik unconformity seem to agree with other lines of evidence indicating a sharp change in grounding-line retreat behavior between ~3.4ka and 3.0ka. If our preferred mechanism for unconformity

20   formation is correct, it records a thinning trend in the ice sheet interior, with a second (1648 ± 92a) undocumented event punctuating a change inan acceleration in thinning inboard of the Siple Coast. The trace of the unconformity indicates that the ice-sheet interior accommodates large-scale grounding-line retreat without dramatic changes in flow orientation. This implies that the ice catchments are less sensitive to more-rapid changes in ice dynamics associated with coastal ice-stream stagnation-reactivation cycles, limiting these cycles' effect on the total flux to the ocean during the time period recorded by the

25   unconformity.

[revised manuscript text omitted]

Woodward, J. and King, E. C.: Radar surveys of the Rutford Ice Stream onset zone, West Antarctica: Indications of flow (in)stability?, Ann. Glaciol., 50(51), 57–62, doi:10.3189/172756409789097469, 2009.

[Figure]

[Figure]

**Figure 1:** (A) Region map showing the modern flow field (Joughin and Tulaczyk, 2002) of Bindschadler (BIS), Kamb (KIS),
5   Whillans (WIS), and Mercer (MIS) ice streams. Mt. Resnik, a subglacial volcanic complex (plotted as a black triangle), sits
adjacent to a tributary of the stagnating Kamb Ice Stream, near the catchment divide between Kamb and Bindschadler Ice
Streams. (B) Map of the subglacial topography at Mt. Resnik (Morse et al., 2002)(Morse et al., 2002). Two ground-based radar
surveys are plotted in black: (i) 1 MHz data collected by the University of Washington in 2004, and (ii) 3 MHz data collected
by St. Olaf College in 2002. (C) Map of modern flow speeds and flow pathways over Mt. Resnik (dashed lines) (Joughin and
10   Tulaczyk, 2002)(Joughin and Tulaczyk, 2002), indicating a dominant flow direction orthogonal to the primary radar survey
orientation. (D) Contoured [20m] surface elevation (Helm et al., 2014)(Helm et al., 2014), plotted with the coverage region

for high resolution DEMs produced as part of this study (blue) calibrated using ICESat altimetry (ground-tracks in orange). data version 34, GLAH06).

[Figure]

**Figure 2**: (A) Fence diagram, indicating the positions of the upstream radar survey (B), the Byrd Ice Core (C), and an example downstream line containing the unconformity of interest (D). (B) The radar profile immediately upstream of Mt. Resnik, highlighting conformable layering. (C) Dated reflectors and their associated uncertainties, plotted as a function of depth at the Byrd ice-core site. Grey bars indicate dated slope breaks in the unconformity, potential indications of historic ice-dynamic changes (discussed in section 3.2). (D) Dated reflectors traced on the 3 MHz ITASE radargram, with ages (in years before 2000 A.D.) labelled. Dotted lines indicate reflectors dated by amplitude and waveform correlation across the unconformity, solid lines are traced continuously from the ice-core site.

[Figure]

**Figure 3**: (A) MODIS Imagery for the region (Haran et al., 2014), showing high reflectivity at the source of the unconformity over Mt. Resnik. Traces of slope-breaks in the unconformity are plotted in orange. (B) False-color Landsat 8 imagery collected over Mt. Resnik, using the near-infrared (band 5), green (band 3), and blue (band 2). (C) Contoured basal topography, with blue-ice areas (inferred from blue to near-infrared reflection intensity ratios > 2) highlighted.

[Figure]

**Figure 4:** Schematic detailing end-member mechanisms for unconformity formation with the distinctive, kinked geometry observed in the radar data: (1) a stationary surface feature orthogonal to flow that mirrors the shape of the unconformity in the subsurface, advecting away and buried with constant velocities, (2) a stationary surface feature orthogonal to flow that is advected and buried in a spatially variable velocity field, with velocity gradients that mirror depth gradients of the structure, or (3), a feature whose boundaries drift with time, with the slope of the unconformity varying as a function of the rate of drift. We believe that mechanism 3 is most consistent with the unconformity geometry, with kink positions corresponding to the dates of ice-dynamic changes in the region.